# Using Brain Activity Patterns to Differentiate Real and Virtual Attended Targets during Augmented Reality Scenarios

**Lisa-Marie Vortmann \***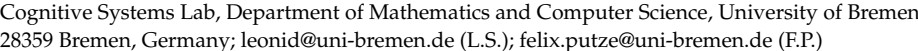**, Leonid Schwenke and Felix Putze**

Cognitive Systems Lab, Department of Mathematics and Computer Science, University of Bremen, 28359 Bremen, Germany; leonid@uni-bremen.de (L.S.); felix.putze@uni-bremen.de (F.P.)
\* Correspondence: vortmann@uni-bremen.de

**Abstract:** Augmented reality is the fusion of virtual components and our real surroundings. The simultaneous visibility of generated and natural objects often requires users to direct their selective attention to a specific target that is either real or virtual. In this study, we investigated whether this target is real or virtual by using machine learning techniques to classify electroencephalographic (EEG) and eye tracking data collected in augmented reality scenarios. A shallow convolutional neural net classified 3 second EEG data windows from 20 participants in a person-dependent manner with an average accuracy above 70% if the testing data and training data came from different trials. This accuracy could be significantly increased to 77% using a multimodal late fusion approach that included the recorded eye tracking data. Person-independent EEG classification was possible above chance level for 6 out of 20 participants. Thus, the reliability of such a brain–computer interface is high enough for it to be treated as a useful input mechanism for augmented reality applications.

**Keywords:** augmented reality; neural networks; eye tracking; classification; attention; EEG





## 1. Introduction

One of the many challenges that our brain is faced with daily is the filtering and processing of vast amounts of information about our surroundings. The input recorded by our auditory, visual, olfactory, gustatory, proprioceptive, and tactile senses is immense at almost any given moment. To survive in a world of sensory overload, we need to give meaning to this available information and focus on the most important aspects of the input. The cognitive process of directing this focus on a selected sensation is the core of attention mechanisms [1]. Subtle differences can still be found between different definitions of "attention" because many processes are still under examination. The meaning, assumptions, and implications about the importance of consciousness, concentration, willingness, allocation of resources, memory, and vigilance are yet to be understood.

As mentioned before, our attention can be directed towards different senses. While at times, our attentional capabilities are best split between several senses equally (i.e., proprioceptive, visual, and auditory while riding a bike in traffic), we sometimes also focus our attention mainly on one sense. Visual attention refers to the conscious and unconscious filtering and selection of visual input [2]. In many cases, this process is linked to gaze behavior, assuming that we direct our eyes at the attended targets (overt attention). Intuitively, the analysis of eye tracking data is often chosen as a means for attention detection. However, the exact gaze point detection requires constant recalibrations to correct for slight movements and sometimes, eye tracking data cannot be recorded with sufficient accuracy if a participant is wearing glasses or has another eye condition. Contrasting overt visual attention, covert visual attention describes directing one's attention to the visual periphery instead of the foveal focus. In this case, eye movement is no indicator for the direction or object of attention [3]. Additionally, attention is more complex than just focusing on specific visual targets or senses. For example, ref. [4] give a detailed taxonomy of internally and externally-directed attention and [5] describe the large-scale

neural network that is associated with exogenous (bottom-up) or endogenous (top-down) attention shifts. In general, it can be seen that different aspects and forms of attention cause different neural activity patterns in the brain [1]. Thus, brain imaging techniques are a good alternative to eye tracking to study and detect complex attention mechanisms, which may not be observed from gaze only.

Electroencephalography (EEG) is one technique that is used to measure such brain wave patterns. The electrical activity is measured with electrodes that are placed on the scalp and recorded by a computer. These recordings can be analyzed and the results can be interpreted as a user input that triggers an action on the connected computer or device. Such a human-computer interaction system is known as a brain–computer interface (BCI) [6]. If a user actively uses the BCI, it can be used for communication (i.e., spelling devices for paralyzed people [7]) or control (i.e., movement of prosthetic limbs [8]). A passive BCI instead makes use of arbitrary brain signals that are not the result of voluntary, purposeful control [9]. When the mental or emotional state of the user changes, the neural activity is unintentionally altered and this information can be extracted and used for adaptations of the connected device (i.e., cognitive fatigue detection in pilots [10]). For such systems to work effectively, the detection of state changes is required to work accurately and in real-time. With its high temporal resolution and the possibility to use a mobile setup, EEG is a good solution for BCIs or cognitive state classification in general.

Especially, if the attentional targets have certain, distinct properties, brain pattern analysis is reliable to classify the current targets of attention, independent of gaze. For example, when the luminance of an attended target flickers in a steady frequency, the same frequency can be observed as a neural response in the brain. This phenomenon is called Steady-State Visually Evoked Potential (SSVEP) and it is a robust detection mechanism for visual attention independent of eye movement to the target and even possible for peripheral vision [11].

Augmented reality (AR) is a relatively new type of user interface that combines real and virtual content. At its core, it is the display of generated information and objects into a natural environment. This merges the near-infinite memory capacity and processing power of computers with human intelligence, information processing, reasoning, and bodily adaptability. The presentation of the virtual content can happen through hand-held devices (i.e., smartphones) or head-mounted displays (HMD, i.e., Microsoft HoloLens, Microsoft, Redmond, Washington, United States ). While the HMD are often see-through for the real surroundings and only project the virtual content on the display, the hand-held devices mostly show a video representation of the environment with the added virtual content (see Section 1.1). This melting of real and virtual information adds to the sensory input and increases the sensory overload. Solving tasks and operating in an environment with a mixture of real and virtual content, therefore, requires sophisticated capabilities to retain attention in order to avoid distraction.

AR introduces two types of distinguishable attention targets in one scene: Real ones and virtual ones. This distinction did not have to be made before, because it is unique to AR but offers interesting information about how users interact with AR and how they process available information. Some AR applications might profit from the information whether the user's attention is on real or virtual information because they can adapt their user interface or behavior for better interaction with the user.

An exemplary use case for the beneficial differentiation between real and virtual objects of attention are industrial augmented reality applications. In [12], the authors identify AR as one of the most important technologies for the Industry 4.0. They analyzed current studies and state that AR can bring value to several industry tasks and sectors: Service, manufacturing, sales and marketing, design, operations, and training. Ref. [13] investigated the usability of AR manuals for factory workers and concluded that visual information is presented more clearly in AR compared to instructions on paper (as a .pdf). For their methodology, they used a structure of simplified text instructions and 2-dimensional graphical symbols. The virtual content is shown concurrently with the

real objects, for example on a screen via a hand-held display. Another AR tool to support workers was suggested by [14]. The tool enables the inspection of three-dimensional models in a real-world context to detect the presence of design or mechanical discrepancies on the final physically assembled product. At the same time, the workers can annotate the three-dimensional models and add their comments or mark any errors. Both of these AR tools, manuals and model annotations, present real and virtual content in close proximity. All instructions or annotations should be placed very accurately and the virtual content must adapt to the real content in the scene and to the current task. If the application was aware of whether the user's attentional focus was on real or virtual content while making annotations, it might add this information explicitly or adapt the presentation of the information for a better communication and collaboration basis across workers on the same project. Depending on the work context, this could further be improved if the virtual content was slightly more transparent and changes in the viewpoint of the worker would result in overlapping real and virtual context. It adds flexibility to the inspection of the model while retaining the high accuracy of the annotations. For the manuals, whenever virtual content overlaps real objects of interest, the application could move or delete the virtual content. The application could also extract the information about the attentional focus as an information source on when to display new instructions. A salient change of the virtual content might distract the user during times of attention to the real object. This could be desired (to prevent errors) or undesired (to not disrupt the process).

Some of the mentioned use case applications could be implemented using just an exact gaze point of the worker. However, as assessed by [15], eye tracking devices usually only have an accuracy of 0.4–0.9 degrees for adults in very fixed settings. They suggest calculating with a 1 degree offset in general which would mean a miscalculation of approximately 1 cm if the target distance from the eyes is 60 cm. Due to the mentioned proximity of real and virtual content, this offset could already lead to a misclassification of the attentional target. Additionally, the systems will mostly be used in a mobile setting, where the eye tracking accuracy further decreases. If the eye tracking device is located in a head-mounted display, frequent recalibrations would be necessary or if the eye tracker was part of a hand-held device, the position of the eye compared to the camera would be subject to constant change, decreasing the accuracy of the gaze point estimation. Lastly, the mentioned overlap of real and virtual content shows another limit of pure gaze point estimation.

As described before, EEG recordings are often used for the classification of cognitive states in general, but also for visual attention. Thus, we considered this a suitable alternative to eye tracking. In [16], the authors discussed several brain recording methods and their usability and application, such as near infrared spectroscopy (NIRS), functional magnetic resonance imaging (fMRI), as well as positron emission tomography (PET). Especially fMRI and the PET were considered not suitable for our study because of the static recording stations that would not allow for the results to be used in portable AR applications. NIRS recordings can be performed in a mobile setup, but the temporal resolution is lower and would not be applicable for real-time systems in the future. Thus, EEG data was chosen as the input for the BCI.

In this work, we performed a study to test how well we can classify attention on real and virtual objects in a controlled augmented reality setting based on EEG and eye tracking data. We implemented a pairs game that had a virtual and a real set of cards and recorded 20 participants while their attentional focus was directed towards the cards. The collected EEG data was classified using a shallow convolutional neural net (CNN) that was built analogously to a Filter Bank Common Spatial Patterns (FBCSP, [17]) feature extraction approach. We compare the results to a simple eye tracking classification approach and a combination of both modalities. We also test the generalizability by analyzing the spectral density of the EEG data for each participant and by testing a person-independent classifier.

The future goal is to work towards a real-time classifier of attention on real and virtual objects that could support pure eye tracking information about the direction of visual attention. The BCI would supply the application with information about the current

attentional state of the user. Depending on the context, as seen in the example, an interface or behavior adaptation might be appropriate to reduce unwanted distractions and improve the usability of the AR system.

### 1.1. Augmented Reality Technology

According to [18], three main characteristics define an AR system: (1) It combines real and virtual content, (2) the interaction with the system happens in real-time, and (3) a reaction and three-dimensional integration of virtual content in the real surroundings takes place. As mentioned before, AR interfaces can be realized through different devices which [19] categorize into three types: Video see-through augmented reality devices record their surroundings and display them through a video with the added virtual content. This version is often used for AR applications on mobile phones and tablets. Optical see-through augmented reality devices instead, use a transparent screen that only displays the virtual content while allowing the user to still see the real surroundings. In Projective AR, the virtual content is projected onto real objects in the environment.

In this work, we use an optical see-through AR device with an HMD that was developed by Microsoft (HoloLens Gen 1, see Figure 1). At the time of the study, it is one of the most advanced devices. All the virtual information is directly projected into the participant's field of view and several cameras scan the surroundings for correct object placement. Interaction with the system is possible through voice control, gestures, and external clicking devices. The participant movement is tracked in relation to the real world, to stick virtual objects to real places, even if they are outside of the field of view.

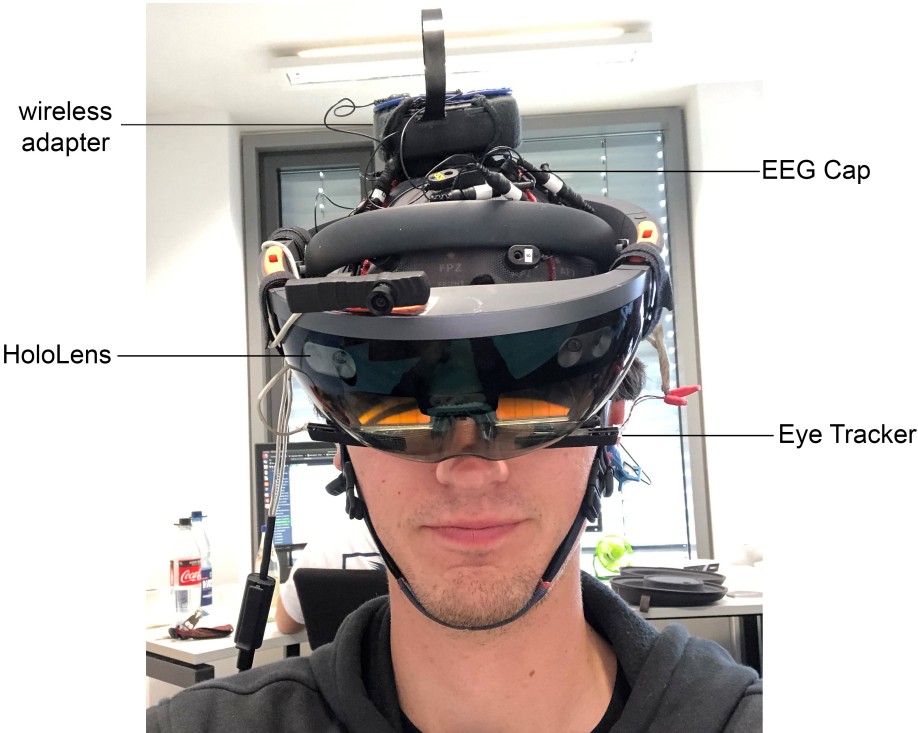

**Figure 1.** Setup of the EEG and the HoloLens during the experimental session.

Currently, AR technology still has its limitations (frame rate, resolution, projection area) and users are usually able to distinguish between real and virtual objects. First of all, virtual objects are usually brighter because they originate from a light source within the device. Additionally, augmented content can withstand physical laws. While the virtual content can be influenced by the real content, the real content can never truly be influenced or changed by virtual content and this flaw can become obvious in different scenarios (i.e., object movements). Another reported projection flaw is that objects can appear to be floating in times when they should not. Further limitations are caused by the

2.4-megapixel display of the device. The 16:9 ratio of the screen offers an HD resolution of 1280 × 720 pixels per eye but restricts the natural field of view. Thus, a virtual object may disappear from the screen, while the natural surrounding where it was placed is still visible for the user (source: https://docs.microsoft.com/en-us/hololens/hololens1-hardware, accessed on 15 April 2021). The display is updated with a refresh rate of 60 Hz.

*1.2. Related Work*

To the best of our knowledge, there are no scientific publications that dealt with the classification of virtual and real objects of attention in EEG-based AR-BCI systems. However, the general use of EEG recordings to assess visual attention was proven to be successful in several studies. Many of these visual attention EEG-studies make use of evoked or event-related potentials, such as Steady-State Visually Evoked Potentials (SSVEP, [20]) or the P300 (positive deflection in voltage after approximately 300 ms, [21]). We will not discuss these studies and their results further in this context, because they are only applicable in specific AR scenarios and applications and require a very specific design.

On the other hand, several research works analyzed general claims about neural activation patterns during visually attentive phases. For example, in [22], the authors report frontoparietal engagement during visual attention tasks. Ref. [23] investigated brain activation patterns for cued visual attention to either the right or the left visual hemifield and found that attention shifts modulate the alpha activity in the contralateral posterior parietal cortex. Sustained attention on a specific target was studied in [24], showing that, in fact, the attention oscillates in a periodic fashion instead of being truly sustained. Ref. [25] found that alpha power increases during times of sensory-input independent tasks compared to sensory-intake tasks. Further, ref. [26] performed visual attention EEG studies using a portable consumer-level EEG headset and proved, that they are good enough to assess event-related potentials of visual attention.

Taking it one step further, EEG data can not only be used to study the neural correlates of attention but it can be used to improve human-machine interaction. Several studies have focused on the modeling of cognitive states for EEG-based BCI systems. Apart from attention, these BCIs can differentiate between levels of mental workload, tiredness, or different emotions [27]. Zhang et al. [28] used EEG data to estimate two states of mental fatigue on a single-trial basis with an accuracy of 91%. A mobile setup was tested in [29], where students' attentiveness was measured in a classroom and correctly classified with an accuracy of 76.82%. Li et al. [30] recognized the potential of EEG for attention estimation and implemented a real-time attention level classifier. The subjects undertook different mental tasks and self-reported their attention in three different levels. The system was able to classify the attentional state in real-time with an accuracy of 57%. Sethi et al. [31] used such an attention classifier for an e-Learning setting to provide feedback to the users on their attentional state. The feedback improved their performance and attention level. To locate user attention and reduce mental workload in video analysis tasks, ref. [32] used eye tracking and EEG data for spatio-temporal event selection. They achieved 91% temporal and 86% spatial accuracy for a static paradigm.

Many BCI systems use gaze as an active input mechanism for controlling an application [33]. The latest AR-headset from Microsoft, the HoloLens 2 even is equipped with in-built eye tracking at a frequency of 30 Hz, and some native applications make use of the gaze point for facilitated user interaction. However, there is more information in the viewing behavior than only the current gaze point. It has been shown that it is possible to classify emotions [34], mental states [35], cognitive deficits [36], and internal thoughts [37] from eye gaze. Annerer-Walcher et al. [38] performed a detailed analysis on several eye tracking features to predict internal from external focus of attention.

One of the technologies that can profit from some of the mentioned BCI systems is augmented reality. The combination of augmented reality and brain–computer interfaces has been of high interest lately. Si-Mohammed et al. [19] published a state-of-the-art summary for AR-BCI systems and described the scope of application. In their work, recent



improvements in the combination of both technologies become apparent, however, they also critically reflect on problems and shortcomings, such as setup and movement artifacts. The problematic combination of two head-mounted systems, like EEG and augmented reality headsets, has also been mentioned by [39]. The correlation of AR with EEG-based BCI-systems has been discussed in [40]. Many AR-BCIs or brain-controlled AR systems make use of specific neural responses such as SSVEP [41] or the P300 [42]. Especially SSVEP-based BCIs are used in many studies to assess the general feasibility of such systems for specific contexts because they are easy to implement and have a straight-forward analysis [43]. For example, ref. [44] suggested controlling the user interface of a medical AR application that enables X-ray vision by adding SSVEP stimuli to normal eye tracking techniques. The feedback of the doctors suggested that such a BCI increases the usability of AR for medical contexts. As mentioned before, these evoked potential studies will not be discussed further for this work. Moreover, we will not discuss any active BCIs where the user explicitly alters his neural activity to evoke an action. Instead, we focus on passive BCIs as described by [45] and how they have been combined with AR.

In 2004, Navarro suggested using AR and Bluetooth as catalysts for a wide use of EEG-based BCIs in his paper "Wearable, wireless brain–computer interfaces in augmented reality environments" [46]. Zao et al. [47] combined EEG and Electrooculography (EOG, measuring eye movements) with AR, suggesting that this type of neurofeedback and neuromonitoring has the potential to improve applications in augmented cognition ranging from feedback-controlled perceptual training to virtual learning and social interactions. Similar to the sample use case from the introduction, ref. [48] replaced the normal input mechanism for an Industry 4.0 inspection tool with a BCI-based input mechanism. Aiming at the restoration of motor control. Chin et al. [49] implemented an AR-BCI that uses a 3D model of a hand to visualize the motor imagery task that the patient is performing. It was found that the BCI-based model is more engaging than conventional visual feedback, even when the majority of the participants are BCI-naive. Barresi et al. [50] claimed that BCI-checked surgical training for users is better than normal training. They combined a BCI with AR and estimated the level of attention. The medical context was also picked up by [51] that used a BCI in combination with AR to assess mental fatigue caused by the visual input. They conclude that the higher workload that is associated with AR may derive from the higher perceived difficulty of tasks in AR. The "Mind–Mirror", implemented and tested by [52], is a direct neurofeedback system that combines visualization of one's own brain in action and a semi-transparent mirror. The virtual model of the brain and current neural activation patterns of the user are displayed on the mirror in the place of the actual brain of the user. This helped the participants to learn to control their mental states and the authors suggest applications ranging from education and training to entertainment. Han et al. [53] argument, that augmented and virtual reality are important technologies for the future of tourism. They suggest a framework to study how these technologies can be used to enhance visitor experiences. In this framework, EEG is used as an experience measure.

The expressed future research goal is to build a real-time BCI for AR that adapts an application's behavior and user interface according to the attentional state of the user. This was implemented for internally-directed (i.e., thought, memory, mental arithmetic) and externally-directed (i.e., visual search, reading) attention in previous work. In [54], we showed that it was possible to classify internal and external user attention in an augmented reality paradigm and in [55], a first real-time attention-aware smart-home system in AR was implemented and tested. It was shown that the usability was improved and the distraction decreased by including system behavior restrictions based on the detected internal or external attention of the user. As input modalities, EEG and eye tracking data was used.

### 1.3. Hypotheses

Based on the related work, the current quality of augmented content, and knowledge about neural processing of visual information, we hypothesize that activity patterns in

the human brain are different for visual attention of real-world objects and virtual objects in augmented reality. This hypothesis is based on the fact that the virtual content is still recognizable as such by the user. Thus, the processing of visual virtual information should evoke a noticeably different neural response than the processing of visual information that is not virtual. Building upon the assumption that there is a detectable difference in the neural response, we hypothesize that state-of-the-art machine learning algorithms should be able to learn this difference and build models for both cases of attentional focus (real and virtual object). Precisely, our main hypothesis is stated as follows:

**Hypothesis 1 (H1).** *In a controlled augmented reality environment, a person-dependent EEG data classification of real or virtual visual attention is possible with an accuracy significantly higher than chance level.*

One major discussion point that is often critical of EEG-based attention classification is the fact that eye tracking data is easier to collect and leads to even better results. Newest augmented reality headsets are even supplied with a built-in eye tracker (i.e., Microsoft's HoloLens 2). This method reaches its limitations for small or overlapping content and is highly dependent on an accurate eye tracker calibration if the gaze point is used to define the current attentional target. Slight movements of the eye tracking device in relation to the eyes would influence the result and constant recalibration would be necessary. Instead, the gaze patterns can be analyzed for differences, as it is often done in eye tracking studies on other cognitive phenomena (see Section 1.2). In our scenario, the viewing behavior for virtual and real objects would have to be noticeably different. We assume, that there are only marginal differences and thus, we hypothesize additionally:

**Hypothesis 2 (H2).** *In a controlled augmented reality environment, basic person-dependent eye tracking data classification of real or virtual visual attention is not significantly more reliable than the person-dependent classification of simultaneously recorded EEG data.*

Assuming that the distinguishable neural activity patterns are evoked by the nature of the virtual representations, they should be similar across participants. Cross-participant EEG pattern recognition for person-independent brain–computer interfaces has been a challenging but desired topic in the field. Individual differences among the participants and users lead to lower classification accuracies compared to models that were trained on person-dependent data. However, we want to analyze whether the models still generalize over participants. We formulate our third hypothesis as follows:

**Hypothesis 3 (H3).** *In a controlled augmented reality environment, a classifier trained on person-independent EEG data can predict real or virtual visual attention of a new participant with an accuracy significantly higher than chance level.*

The focus of this study lies on H1, with H2 and H3 being supporting hypotheses. H2 is taken as a motivation to study this topic and H3 is a preliminary analysis to inspire further thoughts in the direction of training-free real-time BCIs.

## 2. Experimental Design

The major requirements for the experimental task were (1) to ensure retained attention on a real or virtual object over a controlled period of time, (2) a high similarity between the real and virtual trials and their objects beyond the mode of presenting the scene elements, and (3) to avoid strong artifacts in the data. To avoid these artifacts (i.e., caused by movements), we decided to use a controlled and static setting. The retained attention on specific parts of the field of view was achieved by turning the task into a "serious game" [56]. We implemented a card game that follows the idea of the popular game PAIRS in augmented reality. In a game of pairs, the players have a set of cards with distinctive

symbols or pictures on them (i.e., different animals). Each symbol is present exactly twice. The two cards with the same symbol make a pair.

We decided to play the traditional pairs game with adjusted rules in a one-person format, without an opponent. In the beginning, the cards are presented with their symbols facing upwards. The participant is instructed to look at the cards in their positions for 20 s and to remember the positions of as many pairs as possible. This phase is called memory phase. Immediately afterward, in the recall phase, the cards are turned around in the same spot to their empty side and the participant has 20 s to find as many correct pairs as possible by selecting them one by one. When a card is selected, it is turned around to the side with the symbol. If a second card is selected and it does not show the same symbol, as the first card, both cards are turned back around to their empty side. It is considered a mistake but the recall-phase continues. If the two cards match, they stay turned to the side containing the symbol and count as "correctly recalled". In the end, feedback is given about the correct number of pairs that were detected. All participants were asked not to guess randomly but focus on the cards on the memory phase. If several pairs were found correctly within the given time limit of 20 s, it can be assumed that the player was actively paying attention to the cards in the memory phase. Field sizes were chosen randomly to vary the difficulty. The possible fields consisted of $5 \times 2$, $4 \times 3$, $7 \times 2$, or $4 \times 4$ cards. Thus, between 5 and 8 pairs were present in each trial. Following a personal difficulty evaluation, the threshold for the number of pairs that had to be detected correctly to assume a focused Memory-phase was set to 3. Trial difficulty was evenly distributed across the course of the experiment for both virtual and real conditions ($r^2 < 3 * 10^{-3}$ for Pearson correlation between card count and trial position in the experiment).

As mentioned, the game was implemented in AR. With our research goal in mind, we introduced two trial conditions: "Real" and "virtual". In the "real" condition, the sustained attention of the participant is on real objects and in the "virtual" condition, the sustained attention is on virtual objects. Some virtual elements are present in both conditions to optimally simulate a typical augmented reality scenario. These elements are the border of the field, marbles by the side, and a deck of cards. The crucial difference between the two conditions is the playing cards: In the "real" condition, the cards are real physical cards, whereas, in the "virtual" condition, the cards are virtual cards that are displayed by the augmented reality device. The virtual and real cards are identical in size and display the same pairs of matching symbols. The playground is a light grey wooden board in the size of 90 cm $\times$ 40 cm that was placed on a box. The height of this tabletop was the same as the height of the seating of the player. This placement was chosen to give the participant a good overview of the full set of cards. The perspective of the player during the memory phase can be seen in Figure 2 for an example of each condition.

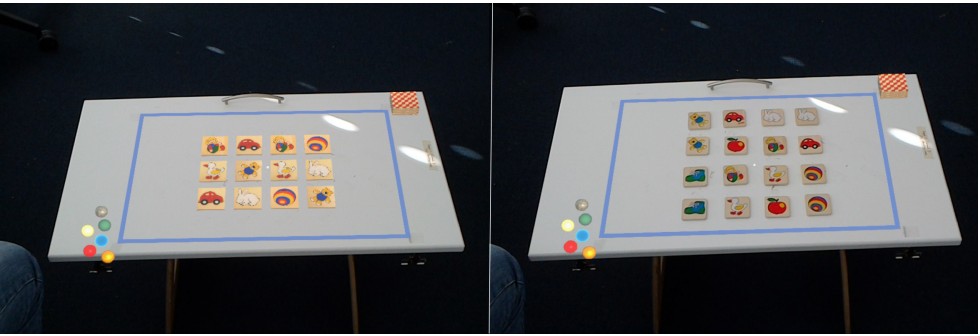

**Figure 2.** Participant's view of the two task conditions. Left: "Virtual" condition with $4 \times 3$ field size; right: "Real" condition with $4 \times 4$ field size; both: Virtual field size border (blue), virtual marbles (lower-left corner), a virtual deck of cards (upper right corner), and a real tabletop (light grey). The two different field sizes were chosen in this example to illustrate the randomly chosen difficulty levels. The field sizes varied from trial to trial.

The only data that was used in the analysis, was the data that was recorded during the memory phase. In this phase, the participants retained their attention either on only real or only virtual content in an augmented reality scenario, while no other actions or cognitive tasks could make a difference between real and virtual trials. This ensures that any difference which is found by the classifier can be tied to the difference in attention on virtual vs. real objects. The data from the recall phase was not used, because during the recall the participant had to perform the action of turning the cards around. This was done manually in the "real" condition. In the "virtual" condition, the participant used the visual "gaze point" that is visible through the augmented reality device and a Bluetooth clicker for selection. Since these performance differences interfere with the pure visual attention on objects and we do not want these differences to be learned by the model, we did not use the data from the recall phase. However, we played the full game to encourage the motivation of the participants and to record the number of correct pairs as a measure of how well the memory phase was performed by the participants.

Another slight difference between the two conditions is the transitions between the phases. In the "virtual" condition all cards were simultaneously presented to start the memory phase and simultaneously turned over after the memory phase to start the recall phase. In the "real" condition, this was not possible because the cards had to be laid out and turned around by hand by the experimenter. To have the same sharp beginning and cutoff of the memory phase in the "real" condition as in the "virtual" condition, a white screen covering the whole visual field of the augmented reality device was added. The white screen covered all the cards so that they were not visible, while the experimenter prepared the field. It was present before the memory phase (while the experimenter laid the cards down) and before the recall phase (when the experimenter turned the cards around to their empty side). This time period was not used during the classification process. The exact procedure of each condition of the task, including the time limits, can be seen in Figure 3.

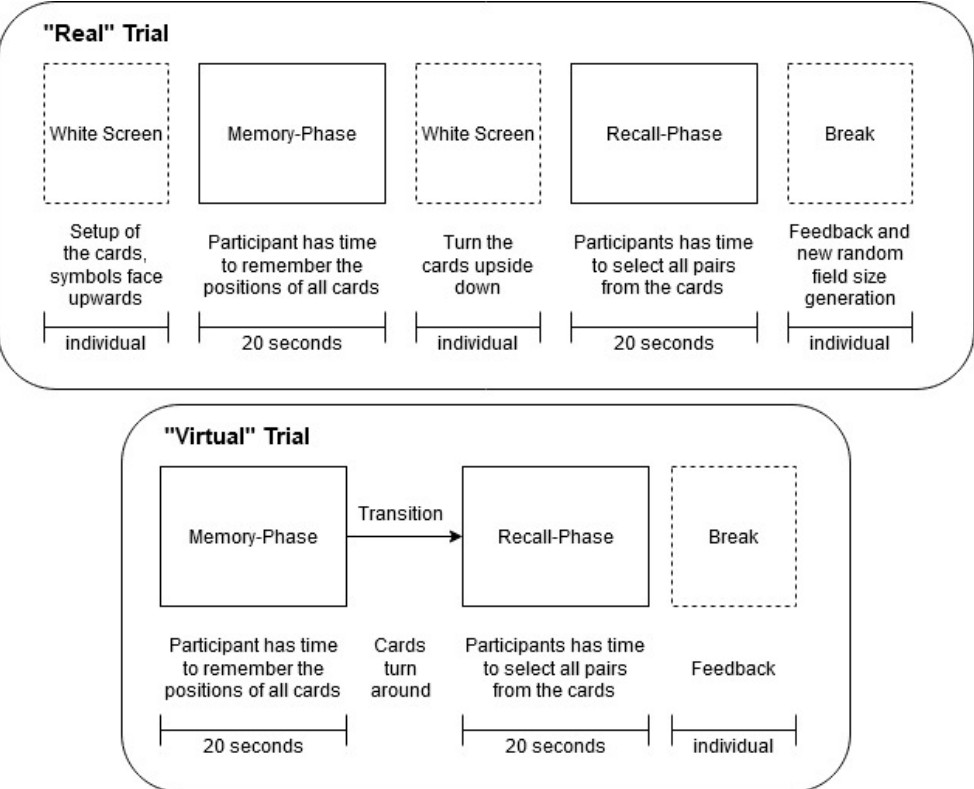

**Figure 3.** Step-by-step procedure of the two task conditions with times and performed actions.

The game was implemented in Unity (version 2018.4.2f1) using the HoloToolKit (version 2017.4.3.0) for compatibility with the Microsoft HoloLens Generation 1.

## 3. Methods

For this within-subject study, we recorded data from 20 participants with normal or corrected to normal vision (age 26.1 ± 7.2; 7 female). Nine of the participants were university students at the time of the recordings (7 in the field of computer science). In a preliminary questionnaire, 11 participants reported that they had experiences with virtual reality and 7 participants reported that they had used an augmented reality device before. After an introduction to the experiment, all participants gave written informed consent to the recording and storing of their data in a completely anonymized fashion. The study was approved by the local ethics committee.

### 3.1. Experiment Session

All experiment sessions took place in the same room of an office building. The room was quiet but not shielded and both sunlight and artificial light were present. A less controlled experiment environment was chosen for a justified comparison to possible real-life applications. The same experimenter attended and instructed all sessions to avoid differences between performance and explanations during the trials. All participants were instructed to come without makeup for better eye tracking results and to not use hair products on the day of the experiment (such as conditioners, gels, sprays, or wax), for a higher EEG calibration accuracy (following common EEG patient guides, i.e., https://www.hss.edu/conditions_eeg-testing-a-patients-guide.asp, accessed on 15 April 2021). The chair and table were set up before the session.

In the beginning, the participants filled out a demographic questionnaire and received a written explanation of the task to ensure that no information was left out. As the first step, the EEG cap was set up and the HoloLens with the eye tracker was adjusted on top of the cap (see Figure 1). For details on the EEG and eye tracking, see Section 3.2. After the setup was completed and the participant was seated in the experiment chair, the eye tracker was calibrated using the manual 3D-marker calibration of Pupil-Labs software Pupil Capture (https://docs.pupil-labs.com/core/software/pupil-capture/#calibration-methods, accessed on 15 April 2021). Unfortunately, eye tracking recordings are only available for 13 participants because of technical problems (no pupil detection possible). All comparisons between eye tracking and EEG data in this study will be based on the 13 participants only.

The experiment was controlled via an experiment terminal by the experimenter. Before the main trials started, each participant executed tutorial trials to get accustomed to the operation of the game. The number of tutorial trials was chosen individually, depending on the feedback of the participant. In total, each participant completed 20 trials of the "real" condition and 20 trials of the "virtual" condition, resulting in a total of 40 trials. These trials were performed in blocks of four trials of the same condition (one trial of each field size). After each block, the condition of the next block was generated randomly with the constraint, that a maximum number of five blocks per condition was possible. The block design was chosen to facilitate and shorten the experimental setup during trials, especially for the "real" trials, where the experimenter was required to set the cards up manually. The field size was also generated randomly for each trial. During the "virtual" trials, errors and correctly recalled pairs were recorded by the application, while for "real" trials, the experimenter noted performance results by hand.

On average, the participants spent less than 60 min with the performance of the experiment. Afterward, all participants answered a questionnaire regarding their perception of the task. The questions compared the perceived difficulty, interaction, and usability of both conditions. Additional free feedback was collected.

Trials, during which technical or environmental problems occurred, were excluded from the analysis. This led to a reduced number of trials for participants 5, 6, 7, 8, and 10

and an overall average of 0.5 deleted trials. The reduced trial number is corrected for in all statistical analyses.

### 3.2. Data Recording

During the main trials, we recorded EEG, eye tracking, and task data with matched timestamps using the Lab Streaming Layer system (LSL https://github.com/sccn/labstreaminglayer, accessed on 15 April 2021). The task data included the beginning and end of trials, as well as their condition, and the results of the recall phase. This data was used for windowing and performance analysis. For the communication between the HoloLens and LSL, we used LSLHoloBridge (https://gitlab.csl.uni-bremen.de/fkroll/LSLHoloBridge, accessed on 15 April 2021). For details on the architecture, see [54].

The EEG data was recorded using a wireless g.Nautilus EEG-headset from g.tec (https://www.gtec.at/product/gnautilus-research/, accessed on 15 April 2021) and 16 gel-based active electrodes. The positions of the electrodes are based on the 10-20-system, covering the whole scalp but adjusted to have minimal interference with the placement of the HoloLens. This resulted in the following placement: Cz, Fp2, F3, Fz, F4, FT7, C3, Fp1, C4, FT8, P3, Pz, P4, PO7, PO8, and Oz. We used a 500 Hz sampling rate during the recordings and impedances were kept below 30 k$\Omega$ (following the suggestions of the manufacturer (http://nbtltd.com/wp-content/uploads/2018/05/grecorderusermanual.pdf, accessed on 15 April 2021)). The data was recorded with a right ear-lobe reference and AFz as the ground electrode. Electrodes whose impedance was above 30 kOhm after the calibration were considered to have insufficient data quality and were excluded for the analysis (on average $1.85 \pm 1.66$ electrodes). In most cases, this was due to a faulty electrode.

Since there is no integrated eye tracker in the HoloLens Generation 1, the PupilLabs binocular eye tracker was mounted to the device. The two cameras that record the eyes are placed under the screen of the head-mounted display and the world-camera is fixated above the screen. The cameras record the eyes with a sampling rate of 120 Hz. Pupil Capture was used to record the gaze position and pupil diameter of the participant. We decided to use basic 2D gaze point coordinates for the analysis in this study, to reflect the eye tracking capabilities of the built-in eye tracker of the HoloLens Generation 2 (as of June 2020).

The pupil diameter reacts to the brightness of the surroundings. Thus, we decided to disregard it for the analysis, because for a generalized claim about augmented reality scenarios, the changing brightness will depend highly on the environment and task and will not be as stable as in our controlled setting. The lighting conditions between the recordings of participants may vary. However, since the eye tracking data is used for person-dependent analysis, this will not be discussed. As reported before, technical difficulties resulted in a reduced set of eye tracking data from only 13 of 20 participants. The average eye tracking accuracy after the calibration was $2.49 \pm 0.51$ degrees.

### 3.3. Analysis

The data analysis, including the preprocessing and the classification, was performed with Python 3.6. All preprocessing steps were kept to a minimum, aiming at a feasible pipeline for a real-time BCI.

Performance statistics of the participants were collected and used to test for the desired focused attention during the task. A very low performance would have resulted in the exclusion of the participant (see Section 3.1). However, that did not happen, and single trials with bad performance results were not excluded.

The EEG data was preprocessed using the MNE toolbox (https://mne.tools/stable/index.html, accessed on 15 April 2021) and following the suggestions of the PREP-pipeline by [57]. The data was band-pass filtered between 3 and 45 Hz using a windowed FIR-filter, excluding the delta-band (1–3 Hz) because it is mainly related to sleep [58]. An additional notch-filter was applied at 50 Hz (power source frequency). The data was re-referenced to average reference. Following the PREP recommendations, we "Detect and interpolate bad channels relative to [this] reference". The bad channels were excluded during the

calibration due to insufficient data quality if the impedance remained higher than 30 kOhm after the calibration (see Section 3.2). We did not visually inspect the EEG recordings for artifacts, neither were any artifacts cleaned automatically. Again, this approach was chosen with a real-time BCI in mind.

Based on the experimental markers, 3 second EEG windows were extracted. This window length was chosen because it allows claims about the feasibility of a real-time BCI in this context. As mentioned in Section 2, only the data from the memory phase will be used for classifier training and testing. The 20-s memory phase was cut into 5 non-overlapping windows (3–6, 6–9, 9–12, 12–15, and 15–18 s after memory phase onset). The first and last seconds of each Phase were left out because the probability for artifacts and missing attentional focus is higher. Afterward, the epochs were baseline corrected using the first 0.5 s of the epochs as the baseline to remove drifts from the data [59].

The cleaned, windowed raw EEG data was used as the input for a shallow convolutional neural network. The network was tested and implemented by [60] and is built following a Filter Bank Common Spatial Pattern (FBCSP) feature extraction pipeline. Based on previous experiments, the learning rate of the model was adjusted to 0.0015 with a weight decay of 0. Their suggested cropped training approach was applied with automatic settings. In all analyses of this study, the neural net was trained for 150 epochs.

### 3.3.1. Trial-Oblivious Approach

For the person-dependent analysis, we first tried a randomized, stratified training-testing split with 30% testing data and repeated the training and testing with 10 random splits for each participant for better accuracy estimation. The splits of the epochs were independent of the trials they belonged to, thus trial-oblivious.

### 3.3.2. Trial-Sensitive Approach

Since we always extract 5 data windows from each trial, the effect of belonging to the same trial within the recording might play an important role for the model. To test and correct for this effect, we additionally used a training-testing split that was trial sensitive. For this approach, all windows from the same trial were either in the training or in the testing data. This trial-sensitive split was also performed ten times, randomly but stratified, with 30% testing data.

In the next step, we analyzed the accuracy that was achieved during the trial-sensitive randomized approach, based on the position of the extracted window within the trial. The question to be answered was, whether any time-frame of the 20-second trial achieved significantly lower or higher classification accuracies than the other extracted windows.

### 3.3.3. Bci-Approach

As the last method for splitting into training and testing data, we chose an approach that represents the real application of a BCI. In the BCI-approach, we maintained the chronological order of trials, i.e., trained the model on the windows of the first trials for each condition and tested the model performance on the windows of the last trials of each condition (70:30 split). If this classifier were used in a real-time BCI, the classifier would be trained on training data that is collected and labeled in a controlled setting before the classifier is used for testing trials in the application.

### 3.3.4. Eye Tracking

For the eye tracking classification, the same windowing was chosen as described for the EEG data. We followed the feature extraction and classification procedure as described in [61], using the scikit-learn toolbox [62]. The calculated features were based on the recorded x and y gaze point coordinates and included information about the outlier quote, fixations, saccades, and gaze velocity and distance. Again, the training and testing split was performed trial-sensitive but randomized 10 times in a stratified manner. The reported accuracies are the average over all ten fitting and testing runs of the LDA. The NN could

not be used for the eye tracking data because it was specifically designed for eye tracking data. The design and evaluation of suitable Neural Nets for the eye tracking classification in the presented task are not within the scope of this work.

We were also interested in the combination of both modalities. In a late fusion approach, we calculated the average accuracy over ten runs for each participant when the EEG prediction was only used if the confidence that was estimated by the NN surpassed a fixed threshold. In all other cases, the eye tracking prediction was taken.

All performed claffifications are summarized in Table 1.

**Table 1.** Description of all performed classification accuracy analyses.

| Approach | Data | Train:Test | Split-Restriction | Classifier |
|---|---|---|---|---|
| Trial-Oblivious | EEG | 70:30 | Stratified | NN |
| Trial-Sensitive | EEG | 70:30 | Windows from the same trial are either all in the training set or all in the test set, Stratified | NN |
| BCI-Approach | EEG | 70:30 | First 70% of the trials of each label are in the training set, last 30% of the trials of each label are in the test set | NN |
| Eye Tracking | ET | 70:30 | Windows from the same trial are either all in the training set or all in the test set, Stratified | LDA |
| Late Fusion | ET EEG | 70:30 | Windows from the same trial are either all in the training set or all in the test set, Stratified | NN Threshold LDA |
| Person-Independent | EEG | Leave-1-out | No data of the test subject is in the training set | NN |

### 3.3.5. Person-Independent Approach

For the third hypothesis, we tested whether the classification of EEG data for this task is possible above chance level for a person-independent classifier. This means that the classifier is never trained on data from the participant whose data is to be classified. The same preprocessed EEG data was taken from all participants and the same neural net as described before was trained in a leave-1-participant-out fashion for all participants. Thus, the model was trained on the data of 19 participants and tested on the remaining full data set of one participant. To add to the understanding of generalizability of the differences in the EEG data, we computed the mean Power Spectral Densities (PSD) for the Alpha (8–14 Hz), Beta (14–30 Hz), Gamma (30–45 Hz) and Theta-band (4–8 Hz) for each electrode of each participant. This results in 16 electrodes $\times$ 4 frequency bands = 64 features that were compared for significant differences. We used MNEs Welch-method to calculate the PSDs. The results for each window were scaled between 0 and 1 based on the minimum and maximum for each feature of each participant individually before computing whether there are significant differences between the "real" and "virtual" conditions if pooled over all subjects. For this analysis, a significance level of $\alpha = 0.001$ was chosen, for a meaningful statement despite the high number of available data windows ($n = 3940$, approximately 200 windows per subject).

### 3.3.6. Evaluation

We evaluated the accuracy, precision, recall, and F1-score for all approaches but we will base our discussion of the results on the accuracy of the classifier. Due to the balanced class distribution, the chancel level for correct window classification is 0.5. Thus, accuracy should represent the quality of the classification well.

In order to rate whether the classification accuracy is significantly higher than chance level, an approach suggested by [63] was followed. Based on the assumption that this

two-class paradigm follows a binomial distribution with $n$ = number of test trials and $p = 0.5$, we can assume that the confidence intervals are given by

$$p \pm \sqrt{\frac{p(1-p)}{n+4}} z_{1-\frac{\alpha}{2}}$$

with $z_{1-\frac{\alpha}{2}}$ being the $1 - \frac{\alpha}{2}$ quantile of the Normal Distribution with a mean of 0 and a standard deviation of 1. The upper border of the interval is calculated for the claim "better than random". If not stated otherwise, we chose the significance level of $\alpha = 0.05$ for all statistical tests in this study. To compare the classification results across training approaches and modalities, paired t-tests were calculated as a significance measure.

## 4. Results

After the evaluation of the performance results, no participant had to be excluded from the analysis. All participants detected more correct pairs than incorrect pairs on average. Summarized over all performed trials by all participants, 5.22 pairs were detected correctly and only 1.5 errors were made. A difference between the conditions can be observed: 77.34% of the trials in the "virtual" condition were complete within the time limit with all pairs, whereas this was the case in 81.35% of the "real" trials. However, due to the different procedures during both conditions, with longer pauses in the "real" condition and higher technical difficulty in the "virtual" condition, no conclusions should be drawn from these results. Importantly, the performance was high enough to assume focused attention during the memory phases.

The results of the questionnaires and individual comments also did not lead to any reasonable exclusion of trials or participants. Overall, the interaction with both conditions was comfortable. It was reported, that the recall phase was harder to perform in the "virtual" condition, but this does not affect the data of the memory phase. Additionally, the questionnaire results show, that the memorization of the cards was neither too hard, nor too easy with an average score of $2.2 \pm 0.95$ for the "virtual" condition and $2.55 \pm 1.1$ for the "real" condition on a scale from 1 = easy to 5 = hard. The free questions on the questionnaires revealed, that the main difference in perceived difficulty is due to the "recall"-phase, where the interaction with the virtual cards was less intuitive. However, since the data of the "recall"-phase is not analyzed, the different ratings are not important. We chose "relatively easy" card layouts because we did not want the participants to develop alternative strategies to remember the card positions. The task aimed to ensure sustained attention to the cards over a longer period of time, which can already be achieved with rather low difficulty. Twelve participants rated the "virtual" condition as preferable and more fun. There was no correlation between classification results and experience with AR.

### 4.1. Person-Dependent Classification

As a significance measurement, the lower border for "significantly better than random" was calculated as described in Section 3.3. For all participants with a complete EEG dataset, the classification accuracy had to surpass 62.25% ($n = 60$). For subject 8, seven trials had to be excluded. The results for this subject were significant if they were better than 64% ($n = 45$). The classification results are considered over both conditions combined because no significant difference between the accuracy of each condition was detected.

The results of the trial-oblivious training sessions for each participant can be seen in the first column of Table 2. The average classification results of the 10 runs that were performed per person are all significantly better than chance. The mean classification accuracy over all participants reached $92.92\% \pm 8.41\%$.

**Table 2.** Overview for trial-oblivious, trial-sensitive, BCI, and person-independent training approaches. Rounded results are shown for all participants individually. Trial-oblivious results are the average accuracy over 10 randomized training-test splits. BCI-approach is the result if the classifier is tested on the last 30% of windows. Person-independent training was performed on the data of the other 19 subjects. n = number of test trials. * Significantly better than random (based on [63] with n as indicated, $p = 0.5$ and $\alpha = 0.05$).

| Participant | Trial-Oblivious | Trial-Sensitive | BCI-Approach | Person-Independent |
|---|---|---|---|---|
| 1 | 0.96 * | 0.69 * | 0.55 | 0.48 |
| 2 | 0.91 * | 0.70 * | 0.63 * | 0.53 |
| 3 | 0.97 * | 0.86 * | 0.83 * | 0.50 |
| 4 | 0.94 * | 0.79 * | 0.55 | 0.63 * |
| 5 | 0.92 * | 0.71 * | 0.65 * | 0.54 |
| 6 | 1.00 * | 0.60 | 0.59 | 0.54 |
| 7 | 1.00 * | 0.71 * | 0.58 | 0.56 |
| 8 | 1.00 * | 0.90 * | 0.89 * | 0.70 * |
| 9 | 0.87 * | 0.65 * | 0.49 | 0.49 |
| 10 | 0.64 * | 0.76 * | 0.89 * | 0.47 |
| 11 | 0.99 * | 0.57 | 0.40 | 0.64 * |
| 12 | 0.97 * | 0.65 * | 0.54 | 0.58 * |
| 13 | 0.96 * | 0.73 * | 0.80 * | 0.62 * |
| 14 | 0.98 * | 0.64 * | 0.54 | 0.48 |
| 15 | 0.86 * | 0.71 * | 0.76 * | 0.51 |
| 16 | 0.80 * | 0.77 * | 0.69 * | 0.51 |
| 17 | 0.97 * | 0.83 * | 0.83 * | 0.59 * |
| 18 | 0.97 * | 0.62 * | 0.58 | 0.53 |
| 19 | 0.92 * | 0.86 * | 0.90 * | 0.53 |
| 20 | 0.99 * | 0.71 * | 0.63 * | 0.52 |
| Mean | 0.93 * | 0.72 * | 0.66 * | 0.54 |
| Std | 0.08 | 0.09 | 0.15 | 0.06 |
| n | 60 | 60 | 60 | 200 |

In comparison, the trial-sensitive approach reached 72.34% ± 8.77% average accuracy over all participants. This is significantly lower than the trial-oblivious approach ($t(19) = 6.9451, p < 0.0001$). When the classifier was not trained on data from the trials it is tested on, it predicts the condition wrong in more cases. On the other hand, the classifier seems to learn which 3-s windows are from the same trial. We conclude that there is a strong temporal dependency in the data and that the trials-sensitive approach represents the difference between attention on real and virtual objects better. The trial-oblivious approach supposedly overestimates the generalizability of the learned model. We will focus our discussion and further results on the trial-sensitive approach. The classification accuracies for each participant are reported in detail in Figure 4. From an individual perspective, the classification was better than random for 17 of the 20 participants. We compared the precision, recall and F1 score for both conditions and the results show the same effects as the accuracy measurements, which was expected because of the equal distribution of the two conditions (see Table 3). Table 4 additionally reports the confusion matrix for both conditions averaged over all 20 participants. In further results, the accuracy of the classification will be reported.

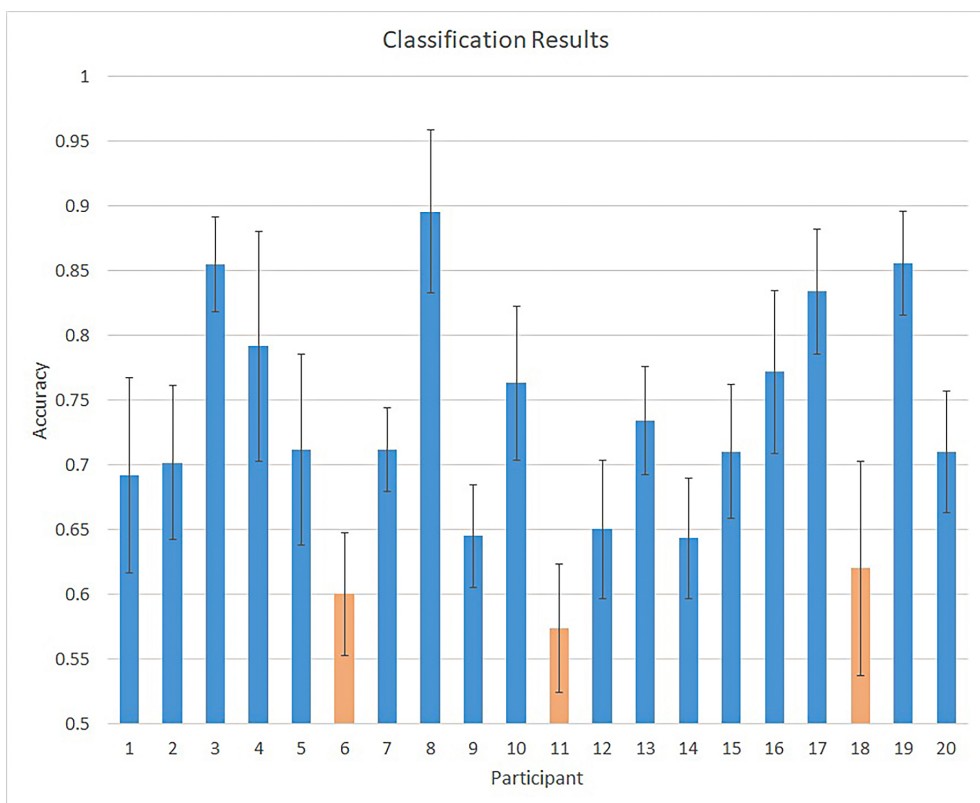

**Figure 4.** Barplots showing the mean classification accuracy and standard deviation of the trial-sensitive randomized approach performed 10 times for each participant. Orange bars are used for participants, where the average classification accuracy was below the calculated threshold for performance better than random.

**Table 3.** Precision, recall and F1 score for real and virtual condition of each participant. ˣ = Participants with an average accuracy that was not significant.

| | **Real** | | | **Virtual** | | |
|---|---|---|---|---|---|---|
| **Participant** | **Precision** | **Recall** | **F1** | **Precision** | **Recall** | **F1** |
| 1 | 0.71 | 0.65 | 0.68 | 0.68 | 0.73 | 0.70 |
| 2 | 0.67 | 0.81 | 0.73 | 0.76 | 0.60 | 0.67 |
| 3 | 0.91 | 0.78 | 0.84 | 0.81 | 0.93 | 0.86 |
| 4 | 0.84 | 0.72 | 0.77 | 0.75 | 0.87 | 0.81 |
| 5 | 0.72 | 0.69 | 0.71 | 0.70 | 0.73 | 0.72 |
| 6 ˣ | 0.61 | 0.57 | 0.59 | 0.59 | 0.63 | 0.61 |
| 7 | 0.72 | 0.69 | 0.71 | 0.70 | 0.73 | 0.72 |
| 8 | 0.85 | 0.94 | 0.89 | 0.94 | 0.86 | 0.90 |
| 9 | 0.64 | 0.67 | 0.65 | 0.65 | 0.62 | 0.64 |
| 10 | 0.74 | 0.75 | 0.75 | 0.78 | 0.77 | 0.78 |
| 11 ˣ | 0.57 | 0.61 | 0.59 | 0.58 | 0.53 | 0.56 |
| 12 | 0.62 | 0.80 | 0.69 | 0.71 | 0.50 | 0.59 |
| 13 | 0.68 | 0.79 | 0.73 | 0.79 | 0.69 | 0.74 |
| 14 | 0.63 | 0.68 | 0.65 | 0.65 | 0.61 | 0.63 |
| 15 | 0.70 | 0.74 | 0.72 | 0.72 | 0.68 | 0.70 |
| 16 | 0.78 | 0.76 | 0.77 | 0.76 | 0.79 | 0.78 |
| 17 | 0.79 | 0.87 | 0.83 | 0.88 | 0.81 | 0.84 |
| 18 ˣ | 0.62 | 0.64 | 0.63 | 0.63 | 0.60 | 0.61 |
| 19 | 0.79 | 0.94 | 0.86 | 0.94 | 0.78 | 0.85 |
| 20 | 0.72 | 0.69 | 0.70 | 0.70 | 0.73 | 0.72 |
| Mean | 0.71 | 0.74 | 0.72 | 0.74 | 0.71 | 0.72 |

**Table 4.** Confusion matrix for the classification accuracy using the Trial-Sensitive approach averaged over all participants (predicted condition × correct condition).

|  | **Real** | **Virtual** |
|---|---|---|
| Real | $0.36 \pm 0.04$ | $0.15 \pm 0.05$ |
| Virtual | $0.13 \pm 0.05$ | $0.36 \pm 0.06$ |

The analysis of the trial-parts showed that the windows extracted from 15–18 s after memory phase onset achieved a significantly lower classification accuracy than the windows extracted from 3–6, 6–9, and 12–15 s and highly significantly lower accuracy than the windows from 9–12 s (see Figure 5).

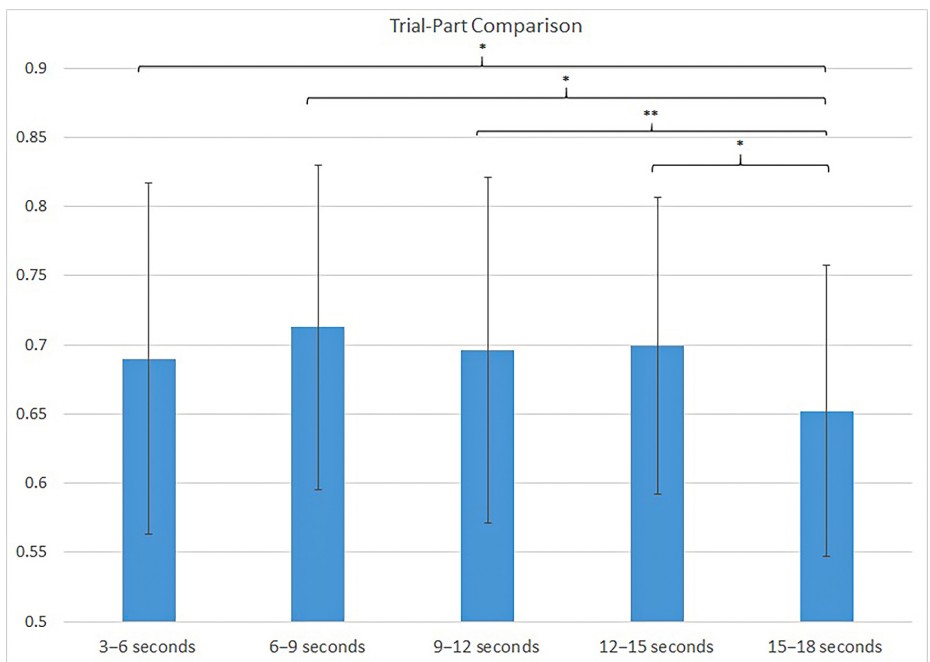

**Figure 5.** The mean classification accuracy and standard error depending on the timing interval of the extracted window within the trial. The results are calculated on all 10 trial-sensitive randomized runs of all 20 participants. Significant differences between the categories are marked. * Significant ($\alpha < 0.05$). ** Highly significant ($\alpha < 0.001$).

If we restrict the trial-sensitive splitting of the training data further by using only the last 30% of trials for testing in the BCI-approach, the average classification accuracy drops to 66.38% ± 14.5%. This overall average classification accuracy is significantly better than random and individually, 11 of the 20 participants had a classification accuracy better than random. For the results per person, see the second column (BCI-approach) of Table 2.

### 4.2. Eye Tracking Classification

Technical problems arose during the recordings of the eye tracking data. For 7 participants, either a successful calibration was not possible because the pupil detection was not stable, or the confidence of the eye tracker decreased dramatically because the lightning conditions changed and again, pupil detection was not possible. This shows that EEG may often be more reliable than ET.

Figure 6 shows a comparison of the 13 participants with full EEG and eye tracking datasets of the study. We compared the mean classification accuracy over 10 classification runs for both datasets for all subjects. The mean accuracy for the eye tracking data was 73.39% ± 7.6% (EEG: 73.63% ± 7.99%). The accuracy difference between the two modalities was not significant in a paired t-test ($t(12) = 0.0784$, $p = 0.9388$). For 5 of the 13 participants, the classifier performance was better for the EEG data, and for 8 participants, it was better

for the eye tracking data. There was no correlation between the results for eye tracking and EEG (Pearson's r = 0.07).

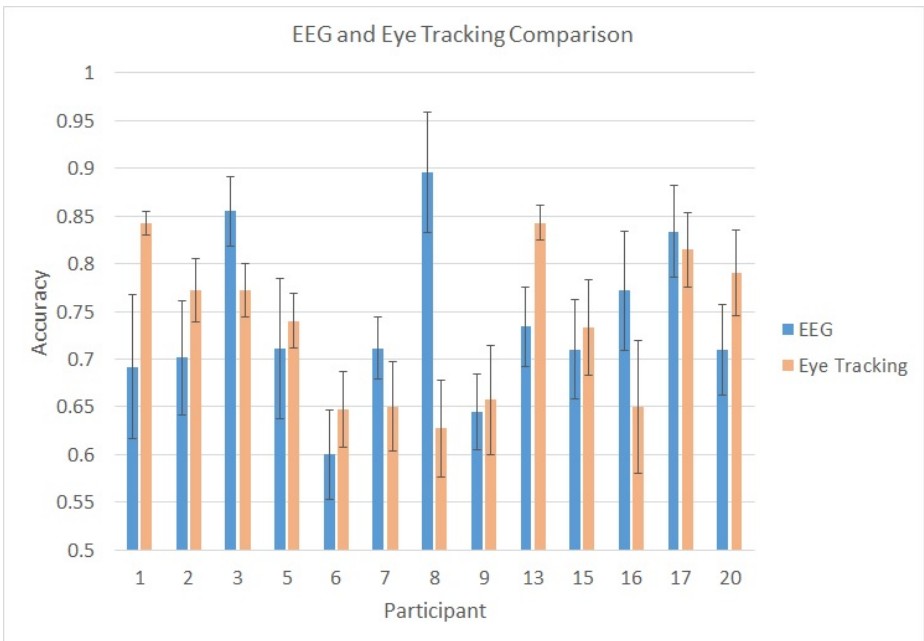

**Figure 6.** Direct comparison of EEG and eye tracking classification results individually. The barplots show the mean and standard deviation over the 10 trial-sensitive randomized runs per participant.

For the late fusion approach that combined both modalities, the average classification accuracy over all 13 subjects increased to 77.47% $\pm$ 8%. This improvement was significant compared to the EEG only result ($t(12) = 3.0114, p = 0.0108$) but not compared to the eye tracking result ($t(12) = 1.6343, p = 0.1281$). On average, 71.2% $\pm$ 10.66% of the predictions were based on the EEG model. Overall, the results improved for 11 of the 13 participants (See Figure 7).

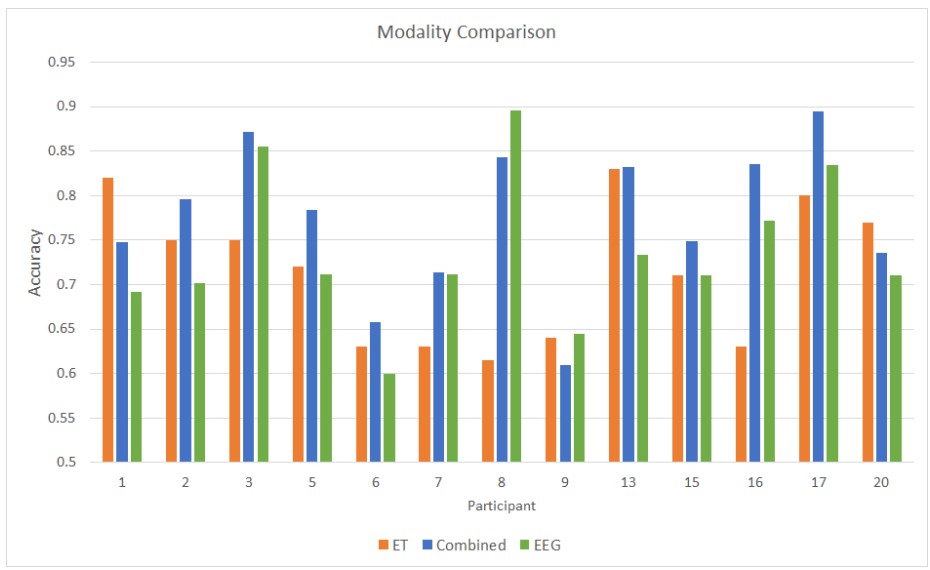

**Figure 7.** Classification accuracies of the single modalities compared to the combined modalities with a late fusion approach per participant.

### 4.3. Person-Independent Classification

For the person-independent EEG classification, the training was performed on the data of 19 subjects and tested on the remaining subject. Following the significance analysis

method of [63], with $n = 200$ and $p = 0.5$, the classification accuracy has to be above 56.8% to be considered better than random. The individual person-independent classification results were better than random for 6 of the 20 participants (see column "Person-Independent" of Table 2). The mean accuracy over all participants reached 54% $\pm$ 6% which is not significantly better than random.

### 4.4. Feature Analysis

To answer the question of what the FBCSP-based neural net learned, different approaches and visualizations were tested. Firstly, we decided to compare the frequency band features of the two task conditions independent of the models. We tested for significant differences if the features are averaged per participant using a t-test for paired samples. With a significance level of $\alpha < 0.05$ for a highly significant difference, three features were selected. A graph for the three features is shown in Figure 8. For the alpha-band, the differences between the conditions were significant for FP2 ($t(19) = -2.618, p = 0.017$) and PO7 ($t(19) = -2.364, p = 0.029$). For FP2, a significant difference was also found for the beta-band ($t(19) = -2.245, p = 0.0369$). characteristic activity in parietal and occipetal regions of the brain is in accordance with the results from [22] on visual attention. The alpha and beta activity in the right frontal region of the brain (FP2) was also linked to attention in [64]. The authors assumed the the association of the brain activity with attentional control via the inhibition of behaviorally irrelevant stimuli.

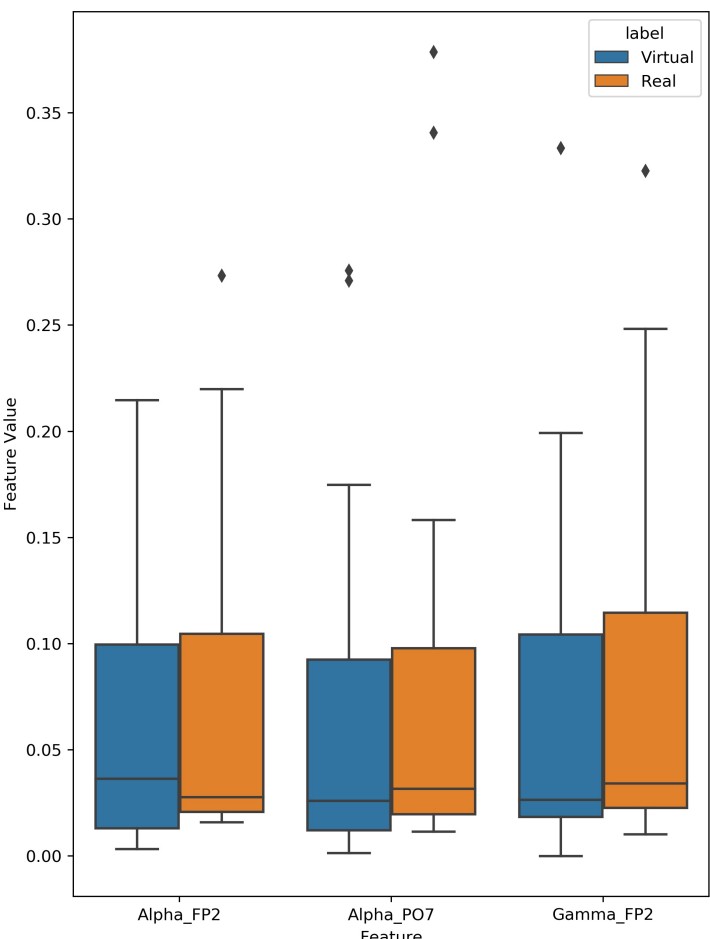

**Figure 8.** Boxplot graph of the features that show significant ($\alpha < 0.05$) differences between the two conditions when averaged per participant. The bar shows the mean value, the box shows the quantiles of the data, and the whiskers extend to show the rest of the distribution. Outliers are indicated with the diamonds.

We also visualized the FBCSP and analyzed what the neural net had learned. However, no common pattern was shared across participants or across person-independent models and the results were not visible in this approach. This is due to the low resolution from only 16 electrodes.

## 5. Discussion

In this work, we used machine learning to classify attentional focus on real and virtual objects in augmented reality. To the best of our knowledge, this is the first work on the subject. We implemented an adjusted pairs game with real and virtual cards and classified 3-s windows of EEG and eye tracking data.

Our first hypothesis H1 was that we would be able to predict the real and virtual targets of attention with an accuracy better than random for person-dependent classifiers. Even after excluding the time effect in the data, the average accuracy over all participants was still significantly higher than chance and individually, 17 of 20 subjects reached a performance significantly better than chance. This reliable prediction supports our hypothesis.

One drawback of BCI systems is that they are not very robust and rarely reach 100% accuracy. Usually, their results worsen with time as movements and environmental factors influence the accuracy [65]. Additionally, it has been claimed that 20% of the participants in a BCI experiment are unable to achieve reliable results because of "BCI-Illiteracy" [66]. This percentage is in line with our results. The reasons for this effect have not yet been determined but could be due to the high interpersonal differences between brain activation patterns [67].

Considering the short time windows of 3 seconds and the sparse positioning of the electrodes, the results are satisfying. We assume that the classification accuracy can be improved by increasing the density of the EEG or optimizing the current placement of the electrodes. Moreover, longer data windows could improve the performance of the classifier. These improvements were not assessed further in this work, because we keep the goal of real-time classification in mind. Further tuning of the setup and preprocessing, as well as a more specialized classification process, could significantly improve the classification accuracies for each individual participant. These steps can be taken if the goal is not a real-time adaptation but for example a post-hoc analysis of how the content of an application was perceived.

Since the distinction is possible based on the EEG data, the question remains what exactly leads to the differences in neural activity patterns for the two conditions. The individual and person-independent feature analyses showed many variances between participants. Possible explanations for distinguishable cognitive user states in this task include aspects of workload, memory, or visual input properties. Depending on the participant, the main perceived differences between the two task conditions might vary. The frequency band analysis sheds some light on the possible distinguishing features: Most features with highly significant differences over all participants were located in parietal and occipital regions of the brain. The occipital cortex contains the visual cortex, which is the primary region for the processing of visual information, while the parietal cortex is activated through sensations and perception and plays an important role in sensory integration, especially for visual input [68–70].

The frequency band that showed the most significant differences was the alpha band. According to [71], alpha-band activity in the parietal and occipital regions is reduced for retained attention to a bright stimulus. As mentioned, the brightness of the real and virtual cards was indeed different and could be an explanation for the results.

It could happen that in the future the AR technology for head-mounted displays improve significantly. It would make it harder to distinguish whether the attention is on a real or a virtual object. However, until then, a differentiation of the two states by our supposed methods is possible and in some cases helpful for the application design.

These initial clues about the differences should be used in further studies on the topic for both setup and additional analysis.

Our second hypothesis (H2) was that the eye tracking classification would not significantly outperform the EEG data classification. Compared to the EEG recordings, the eye tracking recordings had a worse quality and more technical difficulties due to the mobile setup in conjunction with the headset. These problems could be solved in the future with a built-in eye tracker, although it may still suffer from de-calibration, e.g., from head movement. The remaining problems with eye tracking would be the limited spatial resolution of the gaze point and difficulties differentiating objects of attention if real and virtual content is overlapping each other. An EEG-based classification is less prone to a high visual fidelity of the objects. Therefore, the systems would still benefit from the suggested BCI. For our features and classification approaches, H2 proved to be true. It was shown that the classification accuracy of EEG and eye tracking data did not correlate and that in many cases either the classification of the EEG or of the eye tracking data did lead to very high classification accuracies. This suggests that a combination of the two modalities (either in an early or a late fusion approach) would be beneficial for the performance of a classifier. More work would have to be put into the extraction of sophisticated features for this task and a suitable decision function or feature combination could combine the advantages of both approaches. However, finding an optimized classifier was not in the scope of this work.

It could happen that in the future the AR technology for head-mounted displays improves significantly. It would make it harder to distinguish whether the attention is on a real or a virtual object. However, until then, a differentiation of the two states by our supposed methods is possible and in some cases helpful for the application design.

Our third hypothesis (H3) was that a reliable classification is possible even if the classifier is trained person-independently. Person-independence is desirable because it excludes the need for recording training data before the classification in a real-time system. In this study, we achieved results better than chance for only 6 of the 20 participants and the averaged results were not better than chance. Thus, we were not able to prove H3. This suggests large between-person differences that could be due to the task specific solving strategies of the pairs game. On the other hand, we can, at this point, also not rule out that real and virtual attention targets evoke different neural patterns in different participants. Further studies with other experimental setups and a closer focus on feature analysis would be necessary to answer this question.

In this work, the focus during the task design was to have two conditions that are as similar as possible and only differ in the nature of the attended targets. The virtual cards were modeled on the real cards. We only analyzed the time windows during which we were certain that the participants attended the cards. This way, we ensure that the classified differences in the EEG recordings are only caused by this characteristic of the objects. While we eliminate the difference factor, we reduce the generalizability of our results with this task design. For other AR applications, these restrictions are unrealistic and virtual and real objects will certainly differ in size, shape, color, and purpose. We can not make sophisticated claims about how well the classification will work for less controlled environments.

By using only the memory phase and not the recall phase of the task, we eliminate the difference of the user input mechanisms. In the "real" condition, the participants can select and turn cards using their own hands. In the "virtual" condition the cards are turned after being clicked on using a virtual pointer. We made the assumption that the memory phase does not differ for the two conditions, because no interaction with the cards is necessary. While the participants only reported a difference in difficulty for the recall phase, the expectancy of this might have influenced the memory phase. Previous EEG studies have found "premovement" neural acitivty relating to the intention of a movement [72]. However, these appear about 500 ms before the movement and thus, do not interfere with our analyzed time windows.

## 6. Conclusions

Our goal was to perform a first study that tests whether attention on real and virtual objects in AR is represented differently in the brain to an extent that makes machine learning-based classification possible. With our current setup and analysis approach, we were able to prove our first and second hypotheses to be true: Person-dependent classification based on EEG data is possible better than chance and works more reliably than the classification based on eye tracking data. A first feature analysis showed, that the significant differences in neural activity that were detected in our study are consistent with the literature. Even an initial attempt at a person-independent EEG-based classification showed promising results but they were not significant in this study. Thus, we conclude that further research on this topic will attain interesting and useful results for the improvement of augmented reality devices and applications.

*Future Work*

Following the positive results from this study, the next steps will include the implementation of other scenarios to test whether the results were task-dependent. The scenarios will be less static and controlled while improving the setup based on the newly gained knowledge from this study. We know now that the classification is possible and we found plausible features that differ between the conditions for all participants. One question that remained unanswered is whether the classification accuracy decreases for highly experienced users. The participants in this study had some experience but not on a level where we would expect strong differences in the perception. One idea is to compare a group of very experienced AR users with inexperienced users.

In general, the perceived differences between users in AR are an interesting topic and the mentioned aspects of workload, memory, or consciousness of the perception of virtual information are worth further studying.

The improved setup for the next studies will also include adjusted classification processes. The combination of EEG and eye tracking into a multimodal classifier seems promising and even gaze point information could be included as clues for specific applications. Additionally, person-independent classification can be improved and eye tracking data will be included for this. A completely different approach to the classification of attention in this context would be the analysis of the SSVEP, based on the display frequency of the augmented reality device. SSVEP detection with a flickering frequency has been done before [73].

The overall goal is an application that profits from the real-time classification of attention on real and virtual objects in AR.

**Author Contributions:** Conceptualization, L.-M.V. and F.P.; methodology, L.-M.V.; software, L.S.; validation, L.S., L.-M.V., and F.P.; formal analysis, L.-M.V.; data curation, L.S.; writing—original draft preparation, L.-M.V.; writing—review and editing, L.-M.V. and F.P.; visualization, L.-M.V.; supervision, F.P. All authors have read and agreed to the published version of the manuscript.

**Funding:** This research was funded by the "Zentrale Forschungsförderung" of the University of Bremen as part of the project "Attention-driven Interaction Systems in Augmented Reality". Open access was supported by the Open Access Initiative of the University of Bremen and the DFG.

**Institutional Review Board Statement:** The study was conducted according to the guidelines of the Declaration of Helsinki, and approved by the Ethics Committee of the University of Bremen.

**Informed Consent Statement:** Informed consent was obtained from all subjects involved in the study. Written informed consent has been obtained from the patient(s) to publish this paper.

**Data Availability Statement:** The data is available on Open Science Framework at https://osf.io/w bgev/ (accessed on 15 April 2021).

**Conflicts of Interest:** The authors declare no conflict of interest.

## Abbreviations

The following abbreviations are used in this manuscript:

| | |
|---|---|
| AR | Augmented Reality |
| BCI | Brain–Computer Interface |
| CNN | Convolutional Neural Network |
| EEG | Electroencephalography |
| EOG | Electrooculography |
| FBCSP | Filter-Bank Common Spatial Pattern |
| HMD | Head-Mounted Display |
| LSL | Lab Streaming Layer |
| PSD | Power Spectral Density |
| SSVEP | Steady-State Visually Evoked Potential |

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
