# Peer review of "Using Brain Activity Patterns to Differentiate Real and Virtual Attended Targets during Augmented Reality Scenarios"

_information, doi:10.3390/info12060226_

Round 1

Reviewer 1 Report

Undoubtedly, the simultaneous visibility of generated and natural objects often requires users to direct their selective attention to a specific target that is either real or virtual. In this article Authors, investigated whether this target is real or virtual by using machine learning techniques to classify EEG and eye tracking data collected in Augmented Reality scenarios. 

My comments to the article are as follows:

- I propose to eliminate the question from the title. In scientific articles, it should rather not appear in the title. It may, however, be part of a research hypothesis.

- I propose to extend the background to the Introduction article by referring to the correlation of AR technology with BCI, which also uses EEG. For example, refer to the article: Augmented Reality of Technological Environment in Correlation with Brain Computer Interfaces for Control Processes, Recent Advances In Automation, Robotics And Measuring Techniques, Book Series: Advances in Intelligent Systems and Computing, Springer from 2014.

In addition, I recommend to provide you with a broader background in the field of methods of acquisition and analysis of biomedical data collected from the brain. In this regard, reference can be made to, for example, the publications: Methods of Acquisition, Archiving and Biomedical Data Analysis of Brain Functioning, Biomedical Engineering And Neuroscience, Book Series: Advances in Intelligent Systems and Computing, Springer from 2018.

- On Fig. 7 the axis description is missing in the form of indication of units of measurement. Please complete this.

- Conclusions are missing from the article - please add them. I think section 5.1 on Future works should be part of the Conclusions.

Author Response

Thank you for the helpful review of our manuscript. 
Your comments helped us improve the old version. We changed the title, added the suggested references, changed Figure 7, and included a conclusion section.  

Reviewer 2 Report

In the considered manuscript, the authors seek to differentiate attention towards real or virtual objects in AR based on collected EEG data. I think this is an interesting, relevant and novel problem. The paper is written well and has logical and easy to follow structure. The methodology is well justified and the data collection and analysis are sound.
I recommend accepting the paper, but I suggest that the authors make some minor fixes in the Discussion section. Currently this section appears less rigorous and somehow poorly organized, compared to the other parts of the paper. My detailed comments are below.

First of all, I could not find a comprehensive discussion of H3. I believe that the authors should add it, since using a trained model to predict behavior of a new AR user would be indeed an important milestone.

Also, although the authors rightfully explore generalization by participants, they do not seem to mention or discuss generalization of objects. What would be the characteristics of different objects (not cards) for their findings to hold? What would happen if the virtual and real objects are different in size, purpose, etc.? It is probably right to use the exactly similar objects in the experiment to eliminate the difference factor, but I think the above at least deserves discussion.

The final issue is somehow related to the study methodology, as it might introduce an uncontrolled factor. The experimental design is probably fine, but I believe it is still worth discussing or justifying. The authors claim:
618: "It was reported, that the Recall-Phase was harder to perform in the "virtual" condition, but this does not affect the data of the Memory-Phase."
But is it necessarily so? Are there no additional factor affecting brain activity during the Memory-Phase for the participants who know what Recall-Phase follows? E.g. could the Memory-Phase be affected by preparing for physical or non-physical action in the Recall-Phase?

Author Response

Thank you for your helpful review. Your comments helped us improve the old manuscript. 

We added a comprehensive discussion of Hypothesis 3, as well as the other discussion points that were pointed out in your review.